

# Proteomic similarity of the Littorinid snails in the evolutionary context

Arina L. Maltseva[1,*], Marina A. Varfolomeeva[1,*], Arseniy A. Lobov[1,2], Polina Tikanova[1], Marina Panova[1,3], Natalia A. Mikhailova[1,4] and Andrei I. Granovitch[1]

[1] Department of Invertebrate Zoology, St. Petersburg State University, St. Petersburg, Russia
[2] Laboratory of Regenerative Biomedicine, Institute of Cytology Russian Academy of Sciences, St. Petersburg, Russia
[3] Department of Marine Sciences, Tjärnö, University of Gothenburg, Sweden
[4] Centre of Cell Technologies, Institute of Cytology Russian Academy of Sciences, St. Petersburg, Russia
* These authors contributed equally to this work.

Corresponding author
Arina L. Maltseva,
arina.maltseva@spbu.ru

## ABSTRACT

**Background:** The introduction of DNA-based molecular markers made a revolution in biological systematics. However, in cases of very recent divergence events, the neutral divergence may be too slow, and the analysis of adaptive part of the genome is more informative to reconstruct the recent evolutionary history of young species. The advantage of proteomics is its ability to reflect the biochemical machinery of life. It may help both to identify rapidly evolving genes and to interpret their functions.
**Methods:** Here we applied a comparative gel-based proteomic analysis to several species from the gastropod family Littorinidae. Proteomes were clustered to assess differences related to species, geographic location, sex and body part, using data on presence/absence of proteins in samples and data on protein occurrence frequency in samples of different species. Cluster support was assessed using multiscale bootstrap resampling and the stability of clustering—using cluster-wise index of cluster stability. Taxon-specific protein markers were derived using IndVal method. Proteomic trees were compared to consensus phylogenetic tree (based on neutral genetic markers) using estimates of the Robinson–Foulds distance, the Fowlkes–Mallows index and cophenetic correlation.
**Results:** Overall, the DNA-based phylogenetic tree and the proteomic similarity tree had consistent topologies. Further, we observed some interesting deviations of the proteomic littorinid tree from the neutral expectations. (1) There were signs of molecular parallelism in two *Littoraria* species that phylogenetically are quite distant, but live in similar habitats. (2) Proteome divergence was unexpectedly high between very closely related *Littorina fabalis* and *L. obtusata*, possibly reflecting their ecology-driven divergence. (3) Conservative house-keeping proteins were usually identified as markers for cryptic species groups ("saxatilis" and "obtusata" groups in the *Littorina* genus) and for genera (*Littoraria* and *Echinolittorina* species pairs), while metabolic enzymes and stress-related proteins (both potentially adaptively important) were often identified as markers supporting species branches. (4) In all five *Littorina* species British populations were separated from the European mainland populations, possibly reflecting their recent phylogeographic history. Altogether our study shows that proteomic data, when interpreted in the context of DNA-based phylogeny, can bring additional information on the evolutionary history of species.

## INTRODUCTION

Phylogenetic analysis strives to reconstruct the evolutionary history based on variation in heritable traits. At least in some cases, the divergence is considered to be driven by adaptation (*Schluter, 2009*; *Nosil, 2012*). Similar phenotypes (at the molecular levels also) might be formed independently under similar conditions, which is classified as either parallelism or convergence, depending on the degree of genetic relatedness (*Elmer & Meyer, 2011*; *Martin & Orgogozo, 2013*). With respect to this, phenotypic traits alone obviously might be deceptive in phylogenetic studies.

The methodological breakthrough in DNA analysis introduced the molecular markers into phylogenetic studies and led to the development of several quantitative approaches to reconstruct evolutionary processes from DNA variation (*Zuckerkandl & Pauling, 1962*; *Kimura, 1968*; *Felsenstein, 1988*; *Maddison, 1997*). Throughout the "era of DNA-markers" adaptive neutrality has been the central prerequisite (*Edwards, 2009*; *Schlötterer, 2004*; *Patwardhan, Ray & Roy, 2014*). Phylogenetic distance (in terms of divergence time) ought to be measured by random neutral (not adaptive, or affected by genetic hitchhiking, or epistasis, etc.) genetic differentiation. Phylogeny was expected to reflect whole-genome, not locus-specific history since the latter might undergo very strong and fast changes (*Schlötterer, 2004*; *Grover & Sharma, 2016*; *Degnan & Rosenberg, 2009*).

Following the rapid development of sequencing techniques, the next step was taken towards phylogenomics and multi-locus phylogenies (*Edwards, 2009*; *Degnan & Rosenberg, 2009*; *Brito & Edwards, 2009*; *Lemmon & Lemmon, 2013*). The growing number of loci in analyses often provides more confident phylogenetic inferences, but also inevitably includes some loci that are under different types of selection, linked to selected loci or even originated by introgression from other species (*Edwards, 2009*; *Kuhner, Yamato & Felsenstein, 1998*; *Beerli & Felsenstein, 1999*). It is now evident that evolutionary changes affect genome unevenly: there are outlier loci, sometimes forming "islands of speciation", important for fitness and adaptation and consequently displaying outlying pattern of variation (*Luikart et al., 2003*; *Noor & Feder, 2006*; *Michel et al., 2010*; *Feder, Egan & Nosil, 2012*; *Ravinet et al., 2017*). From traditional neutral phylogenies we are moving to the whole-genome view of divergence, incorporating contributions of adaptation, introgression and hybridization.

Yet, even though the DNA-markers approach is able to answer many important evolutionary questions, it is not able to answer all of them. The idea of distinctiveness of species trees from gene trees has a long history since early 1960s (*Cavalli-Sforza, 1964*) until now (*Rosenberg, 2013*). For example, *Degnan & Rosenberg (2006)* showed that, if five or more species are regarded, there always exist branch lengths for a species tree topology, for which gene tree mismatch is more common than concordance between gene and species trees. Even though some efforts were put to minimize misleading effects (*Jewett & Rosenberg, 2012*), "the diversity recovered in our surveys of

DNA sequence evolution within and between species is ultimately an indirect and incomplete window into the history of species" (e.g., due to the different level of complexity of such entities as genes or genomes vs. organisms, population and species) (*Edwards, 2009*). Other reasoning can also be added. Firstly, genomic tools are "too far" from phenotype with its functional aspects, which calls for additional, more physiologically/ biochemically relevant approaches to fill this gap and make feasible functional interpretation of observed genetic differences (rev. e.g., in *Edwards & Batley (2004)* and *Patti, Yanes & Siuzdak (2012)*). Secondly, while phylogeny is a reconstruction of the evolutionary history from the variation in the inherited traits, it is now generally accepted that heritable information in living systems is represented not only by genetic information (*Mameli, 2004*; *Bonduriansky & Day, 2009*; *Danchin et al., 2011*). The incorporation of some non-genetic inheritance phenomena (such as niche construction, developmental bias, epigenetic regulation, etc.) into evolutionary thinking gave birth to the so called "extended synthesis" (*Pigliucci, 2009*; *Pigliucci & Müller, 2010*; *Laland et al., 2015*). Although the need for a revision of the evolutionary theory is being debated (*Laland et al., 2014*), the role of non-genetic inheritance in phenotype evolution can no longer be neglected. One way to incorporate non-genetic inheritance is, again, to include phenotypic data in evolutionary analyses. Thirdly, when analyzing events at shallow evolutionary time scale, the resolution of traditional neutral markers might be insufficient (rev. in *Edwards (2009)*). In the case of ecological speciation (i.e., adaptation-driven sympatric divergence), the neutral whole-genome divergence might be too slow to tell something about a short evolutionary history of a young species. Here again, the neutral genetic/ genomic data need to be supplemented by information on fast evolving parts of a living entity (even though the opposite examples are known as well (*Salzburger, 2018*; *Fang et al., 2018*)). There are multiple examples of using fast-evolving outlier loci to reconstruct recent ecological speciation (*Schluter, 2009*; *Luikart et al., 2003*; *Noor & Feder, 2006*; *Michel et al., 2010*; *Feder, Egan & Nosil, 2012*; *Via, 2009*). However, outliers in a broad sense are not just non-neutrally evolving loci revealed by genome-scans; they underlie proteomic, metabolomics (or morphological, ecological, behavioral, etc.) traits. Analysis of variability in the traits instead of focusing solely on underlying DNA sequences gives us a chance for causative functional explanation of the observed differences, and can bring essentially new information on the evolutionary history of a species.

"The fabric of life is protein-based" (*Karr, 2008*), and the proteome as a molecular phenotype trait is a potentially useful tool for evolutionary studies for several reasons. (1) Protein qualitative/quantitative expression patterns obviously cannot be deduced from genomic data; as the proteome mediates interaction of a genome with the environment, it brings non-redundant information about the physiology of organisms. (2) Protein expression pattern cannot be inferred from transcriptomic data either, as mismatches in quantities of proteins and their corresponding transcripts have been repeatedly detected (*Gygi et al., 1999*; *Greenbaum et al., 2003*; *Mack et al., 2006*; *Waters, Pounds & Thrall, 2006*; *Maier, Güell & Serrano, 2009*). (3) Non-genetic inheritance potentially important for speciation and maintenance of species integrity will affect the expression of proteins (*Danchin et al., 2011*). The earliest taxonomic applications of proteomics date back to the

beginning of 1980s (*Aquadro & Avise, 1981*; *Ohnishi, Kawanishi & Watanabe, 1983*) before they were replaced by DNA sequence analysis. Since then, proteomics studies have been used in a wide range of evolutionary areas, such as ecology, population biology, taxonomy, and evolutionary physiology (*Karr, 2008*; *Dworzanski & Snyder, 2005*; *López, 2005*; *López, 2007*; *Biron et al., 2006*; *Kim et al., 2008*; *Diz, Martínez-Fernández & Rolán-Alvarez, 2012*; *Baer & Millar, 2016*) but still proteomics is far behind genomics in its popularity.

We applied this rationale to an example of the gastropods family Littorinidae (periwinkles), using the proteome as a molecular phenotype trait. These marine intertidal mollusks have been model objects for various studies in evolutionary biology, ecology and adaptation (*Sokolova & Berger, 2000*; *Sokolova & Pörtner, 2001a*; *Sokolova & Pörtner, 2001b*; *Conde-Padín, Caballero & Rolán-Alvarez, 2009*; *Granovitch et al., 2009*; *Johannesson et al., 2010*; *Martínez-Fernández, De la Cadena & Rolán-Alvarez, 2010*; *Panova et al., 2010*; *Ng et al., 2011*; *Panova et al., 2011*; *Panova et al., 2014*; *Canbäck et al., 2012*; *Storey et al., 2013*; *Rolán-Alvarez, Austin & Boulding, 2015*; *García-Souto et al., 2018*; *Maltseva et al., 2016*; *Muraeva et al., 2016*; *Lobov et al., 2018*; *Lobov et al., 2015*). Littorinids provide an opportunity to analyze both closely and more distantly related species from several genera inhabiting contrasting biotopes: *Echinolittorina* and *Littoraria* in tropic and subtropic regions; *Littorina* in moderate and subarctic regions. The set of Northern Atlantics species belongs to the latter genus, forming two subgenera *Littorina* subgen. *Littorina* Férussac, 1822 and *Littorina* subgen. *Neritrema* Récluz, 1869 (*Reid, 1996*). The subgenus *Neritrema* includes two groups of closely related species—the "obtusata"-group (*L. obtusata* (Linnaeus, 1758) and *L. fabalis* (Turton, 1825)) and "saxatilis"-group (*L. saxatilis* (Olivi, 1792), *L. arcana* Hannaford Ellis, 1978 and *L. compressa* Jeffreys, 1865). The sister subgenus to *Littorina Neritrema* is *Littorina Littorina*, containing, among others, *L. littorea* (Linnaeus, 1758). The relationships among the littorinid genera, the subgenera and "obtusata" and "saxatilis" clades are well resolved based both on morphology and several neutral DNA loci (*Reid, 1996*, *1989*; *Reid, Dyal & Williams, 2012*). The hierarchy within the "saxatilis" species group still remains rather questionable, and these three species grouped in different combinations depending on the molecular approach used (*Panova et al., 2014*; *Maltseva et al., 2016*; *Reid, 1996*; *Reid, Dyal & Williams, 2012*; *Crossland et al., 1996*; *Wilding, Grahame & Mill, 2000a*, *2000b*; *Small & Gosling, 2000*). Altogether the littorinids with their well-established molecular phylogeny, some distantly related species living in similar biotopes and other closely related species inhabiting contrasting biotopes, represent a good model for comparative proteomic studies.

The present study aims to assess the utility of proteome variation to infer the evolutionary history of the species under investigation. We do not attempt to infer the true species tree by this quantitative-proteomic approach. Instead, we compare adaptively neutral (represented by neutral DNA-loci) and adaptively non-neutral (evaluated by proteomics) parts of the molecular machinery. Firstly, we compare overall patterns of proteome variation in the littorinids to a well-established phylogeny based on neutral DNA markers. Secondly, we focus on discrepancies between the DNA and proteome trees and

**Table 1 Location of sampling sites, collection seasons, and sample composition.**

| Location | Geographic coordinates | Season | Collected species |
|---|---|---|---|
| Cancale, English Channel, France | 48° 70′ N, −1° 84′ W | May, 2014 | *L. saxatilis, L. arcana, L. compressa, L. obtusata, L. littorea* |
| Tromsø, Barents Sea, Norway | 69° 43′ N, 18° 60′ E | May, 2014 | *L. saxatilis, L. arcana, L. compressa, L. obtusata, L. littorea* |
| Chupa Bay, White Sea, Russia | 66° 29′ N, 33° 68′ W | June, 2014 | *L. saxatilis, L. obtusata* |
| Sheung Pak Nai, New Territories, Hong Kong, China | 22° 27′ N, 113° 58′ E | August, 2014 | *L. ardouiniana, L. melanostoma* |
| Oban, Scotland, UK | 56° 27′ N, 5° 27′ W | April, 2015 | *L. saxatilis, L. arcana, L. compressa, L. obtusata, L. faballis* |
| Eilat, Red Sea, Israel | 29° 30′ N, 34° 54′ E | August, 2015 | *E. millegrana, E. marisrubri* |

attempt to interpret them in the light of the existing knowledge on evolution and ecology of the species and on the function of the involved proteins.

# MATERIALS AND METHODS

## Ethics statement

Periwinkles were collected from wild populations (Table 1). Periwinkles are not endangered or protected species in the study regions, so no special permission for their collection was required. Administrations of the Swire Institute of Marine Science, Hong Kong University (Hong Kong, China), the Arctic University of Norway University of Tromsø (Norway), the Educational and research station "Belomorskaia" of St. Petersburg State University (Russia), the Scotland Association of Marine Science (Oban, Scotland, UK) and Interuniversity Institute for Marine Sciences (Eilat, Israel) were informed about the snails sampling. Collection and animal numbers were approved by these authorities. No special approval could be obtained for sample collection at the site near Cancale (France) because it is not an area of any national park or a private territory, so there was no appropriate authority to apply for permission.

## Sample collection and preparation

Adult individuals of littorinid snails *Littorina littorea* (Linnaeus, 1758), *Littorina fabalis* (Turton, 1825), *Littorina obtusata* (Linnaeus, 1758), *Littorina arcana* Hannaford Ellis, 1978, *Littorina compressa* Jeffreys, 1865, *Littorina saxatilis* (Olivi, 1792), *Littoraria ardouiniana* (Heude, 1885), *Littoraria "melanostoma E Asia"* (is unnamed, see (*Reid, Dyal & Williams, 2010*)), *Echinolittorina millegrana* (Philippi, 1848) and *Echinolittorina marisrubri* Reid, 2007, were collected from wild populations in six geographic locations (Table 1).

After collection animals were transported to the laboratory. The *Littorina* snails from the Northern Atlantic were kept in aerated moist containers at 8 °C and rinsed with salt water once a day. Snails were acclimated to these standard conditions for no less than 2 days and no longer than 1 week before dissection. The *Echinolittorina* and *Littoraria* snails from the Red and East-China Sea coasts, respectively, were kept in dry containers at room temperature and rinsed with salt water once a day.

For sample preparation snails were dissected under MBS-10 binocular microscope at 79-849 magnifications to identify species, sex and possible trematode or other heavy parasitic infection and to separate body parts (head, foot and penis). The mollusks identification was conducted according to the shell form, sculpture and pigmentation, as well as male and female reproductive system anatomy (*Reid, 1986*, *1996*, *2007*; *Granovitch et al., 2008*). The correctness of the tropical species identification was confirmed by Dr. David G. Reid personally. The identification of the species within the *Littorina* "saxatilis"- and "obtusata"-groups (the most challenging part) was performed as described previously (*Reid, 1996*; *Granovitch et al., 2008*). More details can be found in the File S1. Only the reliably identified mature individuals with well-developed reproductive system and free of trematode or other obvious infection were used in further analysis.

Tissues of up to 20 animals (collected from different parts of intertidal area within the same location) were pooled separately for different body parts (foot, head, penis) and sexes; and homogenized in lysis buffer (7 M urea, 2 M thiourea, 4% CHAPS, 25 mm Tris, pH 8.2) using Mixer Mill MM 400 (Retsch). Particles were sedimented by centrifugation 12,000$g$, 15 min, 4 °C and supernatants were frozen at −80 °C until use.

## Proteomic analysis strategy

The "biological noise of the system" due to individual variations and microniche adaptation was leveled by pooling, the short acclimation procedure and including biological replicates into the analysis (rev. in *Karp et al. (2005)*). Samples of somatic tissues of males and females were used as biological replicates, as a previous study showed no significant sex effect at the qualitative level (*Maltseva et al., 2016*). Head and foot tissues used for analysis in this study are less sensitive to transient environmental impacts than for example, gill as was demonstrated earlier in a study on salinity stress (*Muraeva et al., 2016*). Technical replicates (at least two for every sample) were averaged to reduce "technical noise" (*Karp et al., 2005*). Altogether this strategy ensures the analysis of the most stable and reproducible proteome, reducing accidental impacts.

In general, the qualitative strategy of a proteomic study aims for the characterization of the set of proteins present in a sample. This was successfully applied to characterize proteomes of different compartments (organelles, tissues, organs) and certain classes of proteins etc., being an integrative part of structural biology. The quantitative approach estimating proteins abundances is a part of functional biology, and is more informative for studies of physiological changes, adaptive mechanisms, etc. (rev. in *Messana et al. (2013)*). We chose the qualitative approach to compare basic proteomes between species since it is less amenable to transient physiological and environmental impacts. Our approach principally resembles the blind whole-genome assaying methods like AFLP, RFLP, RAPD, etc. in the sense of that the identity of analyzed proteomic markers (protein spots) is unknown. The main difference is taking into account the "working part of the living molecular machinery", that is, the proteome, while abovementioned methods engage DNA-regions, both coding and noncoding. We separated proteins in two-dimensional gel electrophoresis, and ultimately, estimated the number of common and unique (new or altered due to amino acid substitutions, modifications, splicing patterns, etc.) protein

signals in single species or species groups in comparison (more details on our approach and rationale of proteomic vs. genetic data comparison can be found in the File S1).

We followed a gel-based methodology because it has, among others, an advantage of whole-protein analysis and consequently saves combinations of different post-translational modifications as individual signals (*Arentz et al., 2015*). We used differential 2D-electrophoresis (DIGE (*Ünlü, Morgan & Minden, 1997*)) because it allows the simultaneous analysis of up to three samples within the same gel (allowing the reliable detection of even minor structural changes) and provides high sensitivity of protein detection (down to 150–500 pg of a single protein) (*Lilley & Friedman, 2004*). Qualitative gel analysis and spot matching were carried out using PDQuest Advanced 8.0.1 software (BioRad, Hercules, CA, USA) by making whole experiment master gel, allowing for the warping of individual gel scans (*Berth et al., 2007*). Spots were considered reliably detected if they were detected in at least two technical replicates of the same sample, or in one technical replicate in at least two different samples. We have taken into account potential post-translational modifications and/or splice variants by identifying independent signals, because every particular modification/splice form possesses its own features and functions, and appearance of a new form/modification represents an important evolutionary event, comparable to amino acid(s) substitution or even the emergence of a new protein. In total, 796 spots were used for inter-species comparison of proteomes. Protein spots differing between species and available for excision after Coomassie gel staining were subjected to tandem mass spectrometry MS/MS-analysis for identification following standard "bottom up" protocol. Basically, in this method proteins are digested into peptides prior to MS/MS-analysis, molecular masses of resulting peptides and their fragments (after fragmentation in the collision cell of mass-spectrometer during MS/MS-analysis) are determined and compared with MS-spectra predicted from a database by software (see File S1 for more details). The success or failure in protein identification did not influence the results of inter-specific comparisons, and aided only in their interpretation (more details on experimental procedures can be found in File S1).

## Statistical analysis

All analyses were run using R (*R Core Team, 2015*).

### Clustering

Proteomic data were analyzed at two levels. First, data on presence/absence of proteins in samples were used to assess proteomic differences related to geographic location, sex and body part. Second, "consensus" species proteomes were constructed to assess the overall similarity of species proteomes. A protein present in at least one sample of a given species was considered as present in that species. These two datasets were analyzed in the same way: Jaccard dissimilarity index was computed, dendrograms were derived using neighbor joining (NJ) and unweighted pair group method with arithmetic mean (UPGMA) and plotted with dendextend package (*Galili, 2015*). Suitability of NJ vs. UPGMA clustering methods was checked using plots of pairwise distances on a tree vs. original pairwise distances. The both methods yielded comparable results for sample

clustering (File S2); NJ performed slightly better for clustering of consensus species proteomes (File S2).

### Stability of clustering

Branch support was assessed by approximately unbiased (AU) *p*-values using multiscale bootstrap resampling (*Shimodaira, 2004*) with 10,000 iterations in the pvclust package (*Suzuki & Shimodaira, 2006*). This number of iterations ensured accurate estimation of AU *p*-values (their standard errors were less than 0.01). Branch support for midpoint-rooted NJ-based trees was assessed using bootstrap with 1,000 iterations (*Efron, Halloran & Holmes, 1996*) in the ape package (*Paradis & Schliep, 2018*). Stability of UPGMA-based sample clustering was additionally assessed using nonparametric bootstrap with 10,000 iterations in the fpc package (*Hennig, 2007, 2008, 2015*). After the original tree was split into a specified number of clusters, an index of cluster stability was computed cluster-wise as a mean Jaccard similarity ($J$) of the actual cluster to the most similar cluster in a bootstrapped data. $J > 0.75$ corresponds to stable and successfully recovered clusters, while $J < 0.5$ marks "dissolved" clusters with uncertain grouping of objects. In addition, the bootstrap samples allowed to estimate the probability of recovering a particular cluster when the underlying data change.

### Cluster markers

The qualitative differences of the periwinkle proteomes were assessed using indicator species analysis (IndVal) with the indicspecies package (*De Cáceres & Legendre, 2009*). The IndVal method measures the strength of association of specific features with particular groups of samples. It was originally developed in community ecology and has been successfully used for detection of molecular markers (*Feng, Bootsma & McLellan, 2018*). We have run several analyses to detect taxon-specific protein markers using the data on presence/absence of proteins in the samples (for each of the 10 species individually, *Littoraria* (*L. ardouiniana* + *L. melanostoma*), *Echinolittorina* (*E. millegrana* + *E. marisrubri*), the pair *L. arcana* + *L. saxatilis*, the "saxatilis"-species group (*L. arcana* + *L. saxatilis* + *L. compressa*) of the *Littorina* (subgen. *Neritrema*), the "obtusata"-species group (*L. fabalis* + *L. obtusata*) of the *Littorina* (subgen. *Neritrema*), the subgenus *Neritrema* (genus *Littorina*), and genus *Littorina* (subgenera *Neritrema* + *Littorina*)). Group-equalized version of IndVal was used to avoid biases due to unbalanced group sizes (*De Cáceres & Legendre, 2009*). The *p*-values of the IndVal statistic were corrected for false discovery rate in multiple tests (FDR) using Benjamini–Hochberg correction (*Benjamini & Hochberg, 1995*). The proteins with a significant indicator value (FDR-adjusted $p \leq 0.05$) were used to compute specificity (the probability that the sample belongs to the target category given the particular protein has been found), and sensitivity (the probability of finding the protein given the sample belongs to the target category) of cluster markers.

## Phylogenetic analysis

We used sequence data from the previous extensive study on littorinid phylogeny (*Reid, Dyal & Williams, 2012*) to produce a tree only for those species that are included in the present study. Fragments of 28S rRNA, 12S rRNA and cytochrome oxidase subunit I

(COI) were obtained from NCBI; information about sequences used is available in File S1. Sequences were concatenated with different genes allowed to evolve independently. Phylogenetic analysis was performed using Bayesian inference, MrBayes 3.2.6 (*Ronquist & Huelsenbeck, 2003*) with the same parameters as in the original study (*Reid, Dyal & Williams, 2012*): fragments of the three genes were used as a concatenated sequence, where individual genes were unlinked to evolve independently; nucleotide substitution model was GTR + G + I; analysis was performed as two independent runs, five chains in each (four heated and one cold; the first 25% samples from the cold chain were discarded) for 25,000,000 generations with a sample frequency of 1,000, print frequency of 1,000 and diagnostics calculated every 1,000 generations. The convergence between two runs was tested by comparison of statistical parameters in the Tracer Software (http://tree.bio.ed.ac. uk/software/tracer/).

### Comparison of phylogenetic and proteomic trees

Prior to comparison, phylogenetic and NJ-based proteomic trees were made ultrametric using non-negative least squares with the phangorn package (*Schliep, 2011*). The Robinson–Foulds distance (both raw and normalized) between unrooted trees was computed to assess the total number of partitions which are implied by one of the trees, but not the other (*Robinson & Foulds, 1981*). Compositional similarity of proteomic and phylogenetic trees was evaluated using the Fowlkes–Mallows index (*Fowlkes & Mallows, 1983*). Correspondence between inter-species distances on the trees was measured using cophenetic correlation (*Sokal & Rohlf, 1962*). In addition, cophenetic distances on the proteomic and phylogenetic trees were expressed as fractions of maximal tree distance to allow direct comparisons (the matrices of transformed distances are presented in Fig. 1).

## RESULTS

### General trees' topologies

We applied two algorithms for proteomes clustering (NJ and UPGMA) which both gave similar results (Figs. 1 and 2; File S2). We deduced the overall similarity of species proteomes from their clustering based on presence/absence of peptides in consensus proteomes (Fig. 1A; File S2). These results were compared with a phylogenetic tree obtained by Bayesian inference based on three molecular markers used earlier by *Reid, Dyal & Williams (2012)* (Fig. 1B; File S2). The general trees' topologies were similar: the first split was between genera—*Littoraria*, *Echinolittorina* and *Littorina* (however, the genera were split in different order at the proteomic and genetic-based trees); then within *Littorina* subgenera (*L. (Littorina) littorea* and *L. Neritrema* spp. separated), and finally the "saxatilis"-group split from the "obtusata"-group. This was confirmed by the low Robinson–Foulds distance (raw RF = 2, normalized RF = 0.43), high cophenetic correlation coefficient (0.80), and by the distribution of the Fowlkes–Mallows index, which indicated significant compositional similarity for all numbers of clusters, except nine (Fig. 1C). The discrepancy between the trees with nine clusters came from the "saxatilis"—group species cluster, where three species grouped differently (Fig. 1): *L. compressa* was a sister-group to the pair *L. saxatilis*/*L. arcana* in the proteomic tree and *L. saxatilis*

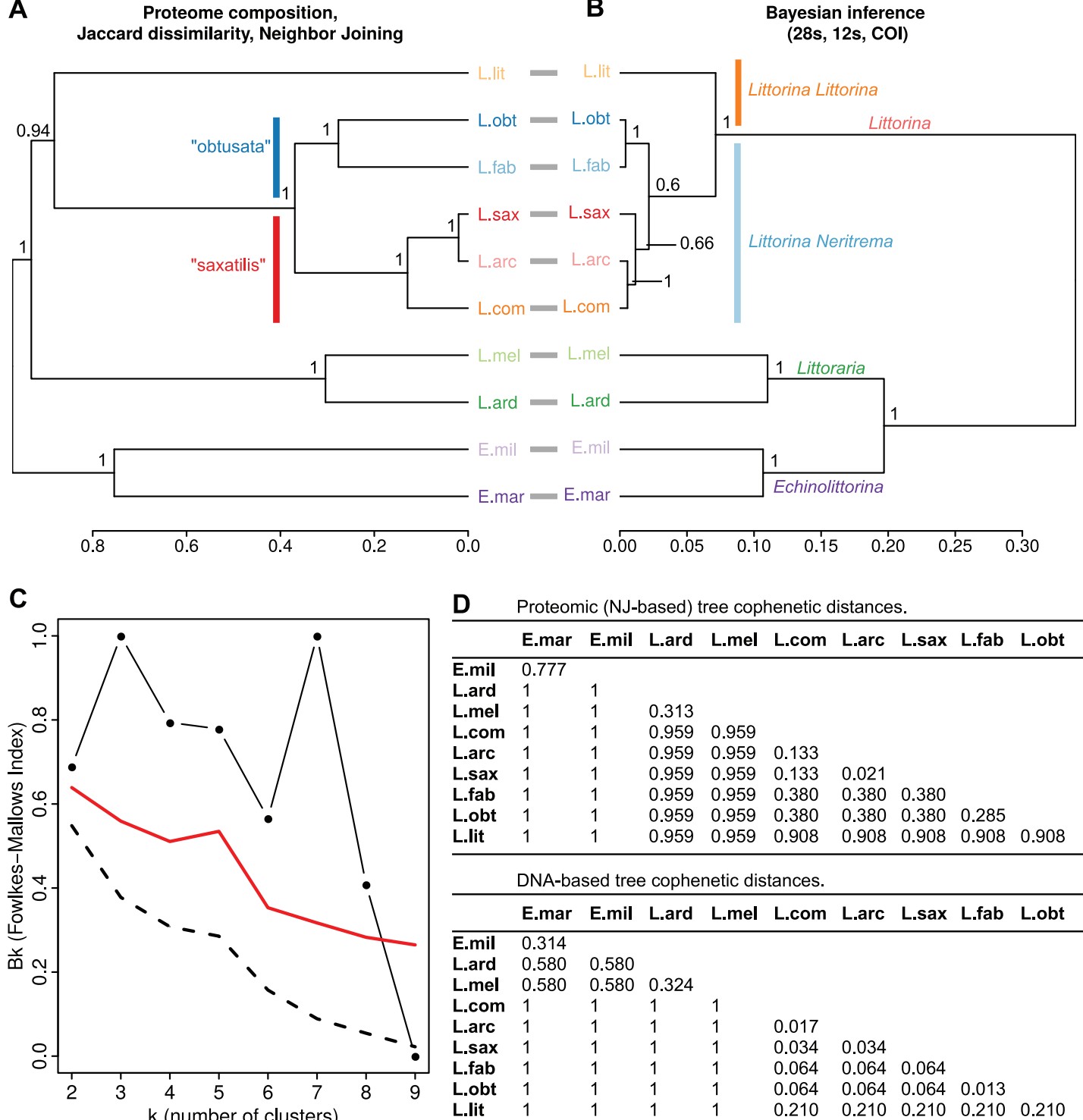

**Figure 1  Interspecies relations within the family *Littorinidae*.** (A) Dendrogram of consensus species proteomes obtained via neighbor joining based on Jaccard dissimilarities of protein occurrence frequency in samples of different species. The bootstrap support values are shown. (B) The molecular phylogeny tree obtained via Bayesian inference using concatenated partial gene sequences from 28S rRNA, 12S rRNA and cytochrome oxidase C subunit I (COI). Support values are posterior probabilities. Prior to comparison, the both trees (A) and (B) were made ultrametric using non-negative least squares. Robinson–Folds distance between unrooted trees was RF = 2 (normalized RF = 0.143). The cophenetic correlation

**Figure 1 (continued)**
between trees (A) and (B) is CC = 0,801; between raw NJ and Bayesian trees is 0.798 (C) Fowlkes–Mallows index comparing dendrograms (A) and (B). Black line with dots shows the change of the compositional similarity of clusters (Bk) with the number of clusters (k). Dashed line indicates Bk values under a null hypothesis of insignificant similarity of cluster' composition in the trees under comparison). Red line depicts threshold values for rejection of the null hypothesis. (D) Matrices of cophenetic distances for the proteomic and DNA-based trees expressed as a percentage of the total tree length. L. lit *Littorina (Littorina) littorea*, L. obt *Littorina (Neritrema) obtusata*, L. fab *Littorina (Neritrema) fabalis*, L. sax *Littorina (Neritrema) saxatilis*, L. arc *Littorina (Neritrema) arcana*, L. com *Littorina (Neritrema) compressa*, L. ard *Littoraria ardouiniana*, L. mel *Littoraria melanostoma*, E. mil *Echinolittorina millegrana*, E. mar *Echinolittorina marisrubri*.          

was a sister taxon to the pair *L. compressa/L. arcana* in the genetic one. Noteworthy, clustering in the proteomic tree was supported by high bootstrap values, unlike low support (0.66) of the "saxatilis"—cluster in the DNA-tree.

## Interspecies cophenetic distances

On the whole, proteomic distances in general exceeded genetic distances (the full matrices of distances, both genetic and proteomic, are available in Fig. 1D). The most robust differences were as follows. Relationships in the "saxatilis" group species cluster: *L. arcana* and *L. compressa* were considered phylogenetically close, but rather distant at the proteomic level (the proteomic and genetic distances between *L. arcana* and *L. compressa* were 0.133 and 0.017, respectively); on the contrary, *L. arcana* was proteomically very similar to *L. saxatilis* in contrast to the genetic distance (the proteomic and genetic distances between *L. arcana* and *L. saxatilis* were 0.021 and 0.034, respectively); the differences were even more profound if UPGMA clusterization algorithm was applied (see File S2). Distances in the "obtusata" group species cluster: *L. fabalis* and *L. obtusata* were identified as phylogenetically very close but strongly diverged at the proteomic level (the proteomic and genetic distances between *L. fabalis* and *L. obtusata* were 0.285 and 0.013, respectively). Their proteomic cophenetic distance is comparable with one between genetically distant *Littoraria ardouiniana* and *L. melanostoma* (Fig. 1A, *Littoraria* species formed common cluster just one step earlier than *L. obtusata/L. fabalis*, Table 2; Table S2). The *Littoraria* genus case. Similarly, proteomic distances were larger than genetic ones for two *Echinolittorina* species and two *Littorina* subgenera (*Littorina* and *Neritrema*). At the same time, two *Littoraria* species showed an unusual outcome: being phylogenetically quite distant, they appeared a bit closer to each other at the proteomic tree (the proteomic and genetic distances between *L. ardouiniana* and *L. melanostoma* were 0.313 and 0.324, respectively), which might be interpreted as a sign of physiological and molecular parallelism.

## Effect of geography, sex and body part

To assess the effect of geography, sex and body part on the similarity of proteome composition, we calculated the extended proteomic tree of samples (Fig. 2; File S2). The general topology of this tree was similar to the consensus one. Almost all species formed clear species-clusters, which supports the potential taxonomic use of proteomics. Nevertheless, there was an exception: the pair *L. saxatilis/L. arcana* formed a mixed cluster, where the factor "body part" (and sometimes "geographic location") affected proteome

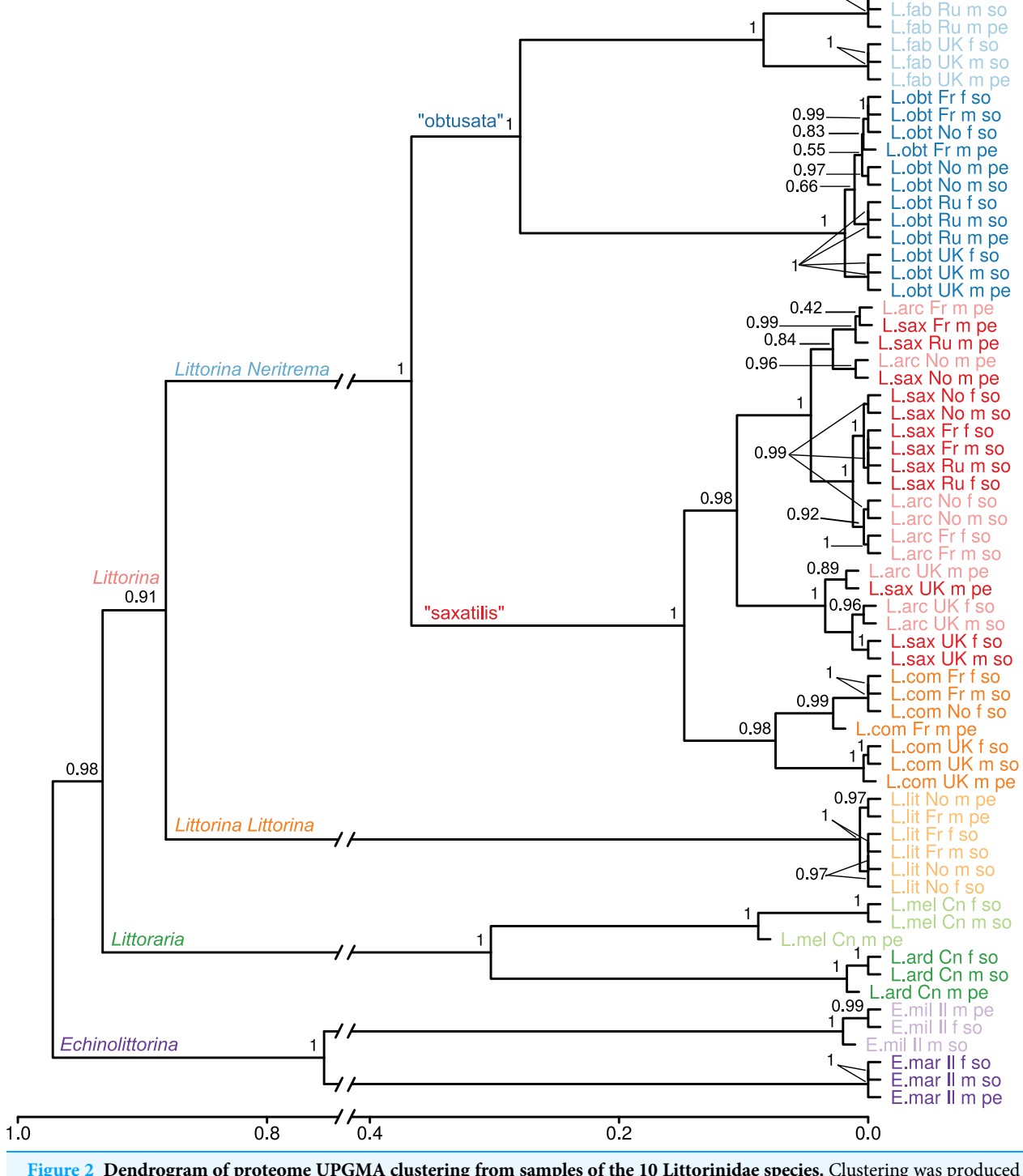

**Figure 2 Dendrogram of proteome UPGMA clustering from samples of the 10 Littorinidae species.** Clustering was produced using unweighted pair group method with arithmetic mean (UPGMA) algorithm based on Jaccard dissimilarity coefficients for the data on presence/absence of proteins in the samples. Sample labels indicate species (L.arc: *Littorina (Neritrema) arcana*; L.comp: *Littorina (Neritrema) compressa*; L.sax: *Littorina (Neritrema) saxatilis*; L.obt: *Littorina (Neritrema) obtusata*; L.fab: *Littorina (Neritrema) fabalis*; L.lit: *Littorina (Littorina) littorea*; L.ard: *Littoraria ardouiniana*; L.mel: *L. melanostoma*; E.mar: *Echinolittorina marisrubri*; E.mil: *E. millegrana*), location (Ru: White Sea, Russia; Fr: English Channel, France; UK: Atlantic coast, Scotland; No: Barents Sea, Norway; Cn: East-China Sea, Hong Kong; Il: Israel), sex (f: female; m: male) and body part (so: foot + head parts; pe: penis). The approximately unbiased bootstrap support values are shown. (neighbor joining-based clustering is presented in File S2).

**Table 2 Composition, protein markers and supporting values of sample clusters.** Selected clusters obtained from analysis of Jaccard dissimilarities for the full set of samples (the full list is presented in Table S1). The first column indicates the composition of a cluster; the second is the number of protein markers, identifying the cluster with maximal specificity and sensitivity; the third is the threshold number of clusters when the cluster of this particular composition appears (from a minimal number of clusters to maximal); the fourth is the recovery probability of the particular composition cluster when it appears for the first time (e.g., the "saxatilis"-group cluster appeared at first during partition into 6 clusters; the recovery probability after this partition was 1; while this cluster persisted through 2 more partitions, its probability dropped to 0.99 and 0.90, respectively).

| Composition of clusters (CC) | Number of markers (NC) | NC for the CC first appearance | Recovery probability |
|---|---|---|---|
| *Littoraria ardouiniana* | 12 | 8 | 0.98 |
| *Littoraria melanostoma* | 24 | 8 | 0.98 |
| *Echinolittorina marisrubri* | 50 | 5 | 0.95 |
| *Echinolittorina millegrana* | 58 | 5 | 0.94 |
| *Littorina Littorina littorea* | 100 | 4 | 1 |
| *Littorina Neritrema fabalis* | 28 | 7 | 0.92 |
| *Littorina Neritrema obtusata* | 28 | 7 | 0.96 |
| *Littorina Neritrema arcana* | 2 | – | – |
| *Littorina Neritrema compressa* | 11 | 9 | 0.99 |
| *Littoraria (L. ardouiniana /+ L. melanostoma)* | 74 | 7 | 0.82 |
| *Echinolittorina (E.millegrana + E.marisrubri)* | 28 | 4 | 1 |
| *Littorina arcana + Littorina saxatilis* | 15 | 9 | 0.89 |
| "saxatilis"-group (*L. arcana + L. saxatilis + L. compressa*) | 33 | 6 | 1 |
| "obtusata" group (*L. fabalis + L. obtusata*) | 12 | 6 | 0.89 |
| *Littorina Neritrema* | 153 | 2 | 1 |
| *Littorina (Neritrema + Littorina)* | 17 | – | – |

stronger than "species". This cannot be due to male species-misidentification, as somatic samples of those males grouped together with unmistakably identified females of these two species. The first within-cluster division splits the UK samples of both species from the mainland ones, suggesting that the geographic variation surpassed the interspecific variability, and a parallel pattern is observed in the proteomes of these two species. Remarkably, distinct UK-clades were also present within the species-clusters of *L. obtusata*, *L. fabalis* and *L. compressa*.

## Cluster stability of samples and cluster markers

To further assess the proteome species-specificity we analyzed the stability of sample clustering (*Hennig, 2007*). Splitting the bootstrapped tree of samples into 10 clusters (the number of species analyzed) resulted mainly in same-species clusters with high recovery probability (Table 2; Table S2), with the exception of the *L. arcana/L. saxatilis* pair which formed two multi-species clusters, corresponding to European mainland and UK locations (both with maximal recover probability). All species branches were supported by the presence of highly reliable (i.e., maximum specificity and sensitivity) protein markers, except for *L. saxatilis* (Table 2), which appeared poorly distinguishable from *L. arcana* at the proteomic level. For *L. arcana* there were only two specific markers. Moreover, these two markers represent two forms of the same enzyme: the peptidyl-prolyl cis-trans isomerase, which additionally reflects the physiological similarity of these two species.

## Markers analysis

We succeeded in the MS/MS-identification of several species (cluster) markers (of 796 protein spots analyzed 128 were successfully annotated, Tables S1 and S3). Expectedly, the most efficient identification was based on a *L. saxatilis* database (LSD) search (*Canbäck et al., 2012*) and gave positive results mostly in the case of *L. saxatilis* and *L. arcana* (~50% efficacy) in comparison to other species (~4–20% efficacy, Table S3). A variety of house-keeping proteins were usually revealed as markers of cryptic groups ("saxatilis" and "obtusata") and genera (*Littoraria* and *Echinolittorina* species pairs), such as protein disulfide isomerase, calreticulin, tropomyosin, troponin T, radular globin, etc. Metabolic enzymes and stress related proteins (both potentially adaptively important), like arginine kinase, fructose-bisphosphate aldolase, malate dehydrogenase, peptidyl-prolyl cis-trans isomerase (PPIase), glutathione-S-transferase, etc., were often identified as species-branch supporting markers (the latter was also the case for *L. saxatilis*/*L. arcana* pair vs. *L. compressa*). The unusual pattern was found for the *Littoraria* species pair, where species with quite long independent evolutionary history matched by core metabolic enzymes (unlike, e.g., "saxatilis" group), but differed by globins (which were common within either the "saxatilis" or "obtusata" group and within the *Echinolittorina* species pair).

## DISCUSSION

In our analysis, the topology of the tree based on the Jaccard dissimilarities of consensus proteomes (Fig. 1) proved to largely agree with the classic morphological (*Reid, 1996*) and the consensus molecular phylogenies (*Reid, Dyal & Williams, 2012*). Moreover, in the tree inferred for location, sex and body part specific samples the strongest differences of proteomes were observed between species: most samples clustered by species (Fig. 2, except for *L. arcana* and *L. saxatilis*, discussed below). This supports the taxonomic utility of proteomic studies, which also was demonstrated in other organisms ranging from bacteria and protists to metazoans (*Dworzanski & Snyder, 2005*; *López, 2005*; *Kim et al., 2008*).

There are however some disagreements in branch length and topology of specific clades between the DNA and proteome trees. Being a kind of molecular phenotype, the proteome embodies both slowly evolving neutral and fast evolving adaptive evolutionary changes. In contrast, classical phylogenetic analyses target neutral DNA loci. Thus, the comparison of these two types of analyses (proteomic vs. genetic molecular markers) can potentially bring some new information about species evolutionary history, as discussed below.

### Similar proteomes in deeply diverged species: a possible case of the whole-proteome convergence/parallelism

In general, the proteomic distances in the trees exceeded the genetic ones. This was rather expectable based on the inclusion of fast evolving proteins; for example, arginine kinases (present as several forms in periwinkles proteomes (*Maltseva et al., 2016*)) were among annotated proteins, and these enzymes are well known for their non-conservative nature (*Uda et al., 2006*). However, there was one remarkable exception from the "large proteomic

distances"-rule. Two deeply diverged *Littoraria* species appeared in the proteomic tree closer to each other than in the genetic one. This result may be interpreted as a molecular parallelism. Although there are many known examples of molecular convergence/parallelism (reviewed in *Storz (2016)*), usually this phenomenon is analyzed at the level of individual proteins of particular families, not the whole proteome. There are strong theoretical arguments for molecular homoplasy resulting from neutral and not adaptation-driven phenomena (*Zou & Zhang, 2015*), for example, a set of structural constraints, limiting protein changes to certain trajectories (reviewed in *Storz (2016)*; *Starr & Thornton (2016)*). Interestingly, in the present study another tropical species pair *E. marisrubri/E. millegrana* (their estimated divergence-time is slightly shorter than that of the *Littoraria* species, ~35 Ma and ~48 Ma, respectively) (*Reid, Dyal & Williams, 2012*) showed no sign of proteomic parallelism, though both species pairs live under extreme conditions of thermal stress, hypoxia and desiccation. Nevertheless, adaptation to a rather special biotope in mangroves, sympatrically inhabited by *L. ardouiniana/L. melanostoma*, could be another possible explanation, as both species are ecologically very similar in terms of host-tree species distribution, diet and activity (*Lee, Williams & Hyde, 2001*; *Lee & Williams, 2002a*). Altogether, the example of the two *Littoraria* species shows that similar proteomes can emerge from divergent genomes.

Interestingly, globin proteins deviate from the observed pattern of whole-proteomic similarity, showing differences between the two *Littoraria* species. This may appear quite puzzling; for comparison, no globin-related difference was observed between the strongly diverged *Echinolittorina* species. A possible explanation for the globin divergence in *Littoraria* species may be differences in their vertical distribution and expected different aerial-aquatic respiration $VO_2$-ratio (*McMahon, 1988*; *Lee & Williams, 2002b*). Similarly, the "obtusata"- and "saxatilis"-groups differ in their preferences to littoral zone level (*Granovitch et al., 2008*, *2004*). Accordingly, they demonstrate different $VO_2$-ratio (*McMahon, 1988*) and different globins, though these species are much more phylogenetically close compared to the *Echinolittorina* species pair.

## Distant proteomes in phylogenetically close species with different habitat preferences

The species pair of the "obtusata" group—*Littorina obtusata* and *L. fabalis* shows a pattern that is opposite to the one described above for *Littoraria*. In the *Littorina* "obtusata" group we observed strongly diverged proteomes, apparently originating from very akin genomes in morphologically similar species. These two species are phylogenetically very close (closer than any other in the Littorinidae family; estimated divergence time is less 1 Ma) (*Reid, Dyal & Williams, 2012*) and were shown to share some mitochondrial haplotypes (*Kemppainen et al., 2009*) (and there are signs of limited gene flow between these two species in Portugal (*Costa et al., in press*)). Here, the comparative proteomic analysis revealed robust differences between *L. obtusata* and *L. fabalis*. The distance between them highly exceeds those within "saxatilis"-group. The discrepancy between genetic and proteomic similarities can be an indirect evidence of ecological speciation. *L. obtusata* and *L. fabalis* demonstrate different preferences in their microhabitat

distribution: *L. fabalis* occupies lower intertidal and subtidal levels, while *L. obtusata* prefers the middle littoral zone; at those levels, they are associated with different fucoids (*Reid, 1996*; *Granovitch et al., 2008*, *2004*). One can hypothesize that partitioning of ecological niches drove divergence at the physiological level, finally reflected in very different proteome patterns in these two species. The observed striking proteomic differences were achieved by these two species in relatively short evolutionary time, which implies a high evolvability of the system. However, the exact nature of proteomic differences is unknown (whether it is related to genes duplication, amino acid substitution, gene expression pattern shift, alternative splicing or some posttranslational modification), and at least part of the differences can be due to phenotypic plasticity. Thus, these species present an interesting system to further investigate mechanisms of rapid proteome divergence.

## Distant proteomes in phylogenetically close species with similar habitat preferences

The third example, from the *Littorina* "saxatilis" group, suggests that different proteomes can be generated by similar genomes even if corresponding species populate very similar biotope. The three species in this group (*L. arcana*, *L. compressa* and *L. saxatilis*) have a high degree of whole genome similarity, as was demonstrated by comparative genomic hybridization (*Panova et al., 2014*). Earlier molecular phylogenies were inconclusive regarding these species relationships (*Canbäck et al., 2012*; *Reid, 1996*; *Reid, Dyal & Williams, 2012*) and different molecular markers clustered the three species of the "saxatilis" group in all possible combinations (*Panova et al., 2014*; *Reid, Dyal & Williams, 2012*; *Wilding, Grahame & Mill, 2000a*, *2000b*; *Small & Gosling, 2000*; *Knight & Ward, 1991*), which is one line of evidence for their close phylogenetic proximity and a recent evolutionary divergence (divergence time is estimated as ~1–1.5 Ma) (*Reid, Dyal & Williams, 2012*).

In our proteomic tree, the first divergence event splits *L. compressa* from a tight alliance of *L. saxatilis*/*L. arcana* (Fig. 1A). We found 11 species-specific markers for *L. compressa* and a similar number of markers specific for the pair *L. saxatilis*/*L. arcana*, but only two markers specific for *L. arcana* and none for *L. saxatilis* (Table 2). These results are congruent with the fact that both *L. saxatilis* and *L. arcana* demonstrate parallel proteomic shifts along the vertical shore gradient while *L. compressa* does not (*Maltseva et al., 2016*). The separation of *L. compressa* at the proteomic level partially resembles the foregoing case of *L. fabalis* and *L. obtusata* in a sense that it occurs between very closely related species. However, unlike the latter pair, the proteome divergence is not easily explained by ecology in this case. Both *L. arcana* and *L. compressa* are very similar to *L. saxatilis*: their ranges and habitat distribution overlap (*Reid, 1996*; *Granovitch et al., 2013*). Although *L. compressa* was not recorded from fucoid-free substrates, unlike two other species, it invariably lives in sympatry with *L. saxatilis*. All three species are often found together on the same fucoids and rocks within an intertidal area, and there are so far no data suggesting stronger differences in ecological preferences between *L. compressa* and *L. saxatilis* than between *L. saxatilis* and *L. arcana*. It is still possible that *L. compressa*

occupies very specific microhabitats to be elucidated in future studies. Alternatively, some hypothetical switch could have occurred in the molecular machinery functioning of *L. compressa*. The last explanation suggests that speciation events could be related not only to environmental shifts or niche expansion but also to changes in an organism's patterns of interaction with the same (or very similar) environment. Among protein markers specific to *L. compressa* we identified arginine kinase, fructose-bisphosphate aldolase and glutathione-s-transferase. These enzymes are potentially related to stress adaptation (e.g., isolation-related hypoxia during low tide; in details discussed in *Maltseva et al. (2016)*); and such changes are compatible with any of the two proposed explanations.

## Strinkingly similar proteomes in phylogenetically close species

In contrast to the *L. compressa* case discussed above, the pair *L. saxatilis*/*L. arcana* showed high proteome similarity. In general, this similarity is expected pattern given the short divergence time and overlapping habitats. What is surprising is that both "geographic location" and "body part" showed stronger impacts on the proteome than "species"; that is, the first split occurs between UK and other regions (further discussed below), then between "foot + head" and "penis" samples, and then between the species (Fig. 2). The misidentification of morphologically similar males of *L. saxatilis* and *L. arcana* could not explain this pattern, because female "foot + head" samples cluster together with male "foot + head" samples; and females of the two species are unambiguously distinguishable.

*L. saxatilis* and *L. arcana* have strikingly similar proteomes (at least in adults and the analyzed body parts), even though they have different reproductive strategies (*L. arcana* is oviparous while *L. saxatilis* is ovoviviparous) and demonstrate some differences in distribution, morphology and genetics (*Panova et al., 2014*; *Reid, Dyal & Williams, 2012*; *Wilding, Grahame & Mill, 2000a*; *Granovitch et al., 2008*; *Knight & Ward, 1991*; *Mikhailova et al., 2009*). Given their genomic, morphological, ecological and now even proteomic similarity, this pair of species raises intriguing evolutionary questions. (1) What mechanisms maintain this similarity across their range? (2) What kind of barrier separates them from each other given indirect evidence of gene flow between them (*Granovitch et al., 2013*; *Mikhailova et al., 2009*)? (3) What evolutionary forces drove their divergence in the first place?

Only two species-specific protein markers were detected in our analysis for *L. arcana* (and none for *L. saxatilis*). Both *L. arcana*-specific markers (and one marker of *L. compressa*) proved to belong to PPIase class of enzymes, the cyclophilins (Cyp) family. These ubiquitously distributed enzymes are known to facilitate folding of certain proteins and thus regulate diverse cellular processes (rev. in *Galat & Metcalfe (1995)*; *Fischer & Aumüller (2003)*; *Lu et al. (2007)*). In mollusks, Cyps are expressed in hemocytes and gonads and involved in immune response (*Song et al., 2009*; *Ji et al., 2013*). They also participate in the processes of biomineralization during shell and radula formation (*Jackson et al., 2009*; *Marie et al., 2013*; *Nemoto et al., 2012*). These enzymes are believed to be secreted to spatially organize matrix proteins, and thus to control crystallization. Hypothetically, structural differences in Cyps between closely related species could cause tiny changes in this process resulting in interspecies radula variation, which was

documented for *L. arcana* and *L. compressa* (*Reid, 1996*). In our analyses we used head tissues including odontophore and radular invagination, where radula synthesis occurs. Moreover, Cyp of *L. saxatilis* is expected to be catalytically active as there are no substitutions in catalytically important amino acids (based on Cyp-transcript sequence present in LSD transcriptome (*Canbäck et al., 2012*)). Further studies are needed to examine the precise compartment of expression and functioning of the identified Cyps.

## Legacy of post-glacial history in proteome divergence

Overall, samples from different localities grouped by species implying a tight link between genome and proteome and a minor influence of physical factors, locality or season. Nevertheless, the same clustering pattern repetitively appeared within every *Neritrema* species: UK samples formed a distinct clade (Fig. 2), while continental samples clustered more often according to the tissue type. In the pair *L. arcana/L. saxatilis* separation between UK and continental samples even precedes the separation of the species (Fig. 2; Table S2). *L. saxatilis* UK populations have been previously shown genetically distinct from the mainland using mitochondrial markers (*Panova et al., 2011*; *Doellman et al., 2011*) and it has been suggested that UK *L. saxatilis* persisted in the UK within some local refugium during the last glacial maximum with following postglacial expansion on the British Islands, but not to the mainland. European mainland populations of *L. saxatilis* were presumably recolonized from other refugia than the UK (*Panova et al., 2011*; *Doellman et al., 2011*). The existence of a UK refugium has been also suggested for *L. compressa* and *L. arcana* (*Doellman et al., 2011*). Our findings are well in line with such scenario. Moreover, both "obtusata" species have possibly also survived in a hypothetical UK refugium, as a separate UK-clade was also detected in these two species. Recently, a new phylogeographic study suggested the UK as a possible glacial refugium for *L. fabalis* and *L. obtusata* (*Sotelo et al., in press*). Overall, we observed a pattern of geographic isolation at the proteome level, which is very similar to effect of the recent post-glacial history observed in mtDNA. It would be interesting to test this effect further by proteomic analyses of other populations known to have survived in separate glacial refugia, such as *L. saxatilis*, *L. obtusata* and *L. fabalis* in the Iberian peninsula, or *L. saxatilis* and *L. obtusata* in North America.

## Qualitative proteomics is not reliable for intergenera analysis

When qualitative data of different genera are put into comparative analysis, proteomics cannot resolve their relationship (unlike the interspecies analysis). This occurs because too many proteins are already changed and too few are conserved, yielding a low number of common "anchoring" proteins between strongly diverged genera and poor grouping of the genera with comparably high binary dissimilarity. This is well illustrated by short genera branches relative to species ones (Figs. 1 and 2). Thus, although in our study genera clustered differently than in DNA-markers-based tree, this result is not sufficient to claim a conflict between the two approaches in this case.

## CONCLUSIONS

Here we evaluated the usefulness of the proteomic approach in comparative evolutionary studies. Proteomics has an advantage of the ability to reflect the biochemical machinery of life in comparison to morphological traits. In addition, it records the features undetected by genomic approaches, such as stable patterns of molecular functioning (including those maintained by non-genetic mechanisms, but crucial for typical species physiology). This allows to easily reveal potentially adaptive differences (like fast evolving proteins) in diverging taxa. Owing to incorporation of both slowly and fast evolving proteins, proteomes may be a basis for higher resolution reconstruction than neutral DNA-markers at short evolutionary scales. For the same reason proteome-based analysis is less informative for genera and higher rank taxa, because they might have accumulated too many changed proteins. Our data demonstrate that, although there is a clear connection between the proteome and genome, proteomic and neutral DNA-markers-based evolutionary reconstructions can either agree or contradict each other, depending on whether adaptive processes have been involved in the recent history of the taxa in comparison. Any of the particular outcomes can be informative for the interpretation of species relationships, their evolutionary histories, and the causal and functional background of their evolution.

## ACKNOWLEDGEMENTS

The opportunities for 2D DIGE and LC-MS were provided by the Resource Center "Molecular and Cell Technologies" of St. Petersburg State University; the opportunities for laboratory maintenance of experimental animals were provided by the Resource Center "Observatory of environmental safety" of St. Petersburg State University; the opportunities of snails collection at the White Sea coast were provided by Educational and research station "Belomorskaia" of St. Petersburg State University. We thank David G. Reid for his kind help with species identifications of the *Littoraria* and *Echinolittorina* snails, Anna Gonchar for proofreading and language editing; two anonymous reviewers for valuable and very helpful comments on the manuscript.

### Funding

This research was funded by Russian Foundation for Basic Research Grants Number 18-54-20001 and 18-34-00873 and by St. Petersburg State University Grant Number 0.40.491.2017. The funders had no role in study design, data collection and analysis, decision to publish, or preparation of the manuscript.

### Grant Disclosures

The following grant information was disclosed by the authors:
Russian Foundation for Basic Research: 18-54-20001 and 18-34-00873.
St. Petersburg State University: 0.40.491.2017.

## Competing Interests

The authors declare that they have no competing interests.

## Author Contributions

- Arina L. Maltseva conceived and designed the experiments, performed the experiments, analyzed the data, prepared figures and/or tables, authored or reviewed drafts of the paper, funding aquisition, and approved the final draft.
- Marina A. Varfolomeeva conceived and designed the experiments, analyzed the data, prepared figures and/or tables, authored or reviewed drafts of the paper, and approved the final draft.
- Arseniy A. Lobov performed the experiments, authored or reviewed drafts of the paper, and approved the final draft.
- Polina Tikanova performed the experiments, authored or reviewed drafts of the paper, and approved the final draft.
- Marina Panova analyzed the data, authored or reviewed drafts of the paper, and approved the final draft.
- Natalia A. Mikhailova conceived and designed the experiments, authored or reviewed drafts of the paper, and approved the final draft.
- Andrei I. Granovitch conceived and designed the experiments, analyzed the data, authored or reviewed drafts of the paper, administration of the project, funding aquisition, and approved the final draft.

## Field Study Permissions

The following information was supplied relating to field study approvals (i.e., approving body and any Reference Numbers):

Periwinkles are not the endangered or protected species, so no special permission for their collection was required. Administrations of the Swire Institute of Marine Science, Hong Kong University (Hong Kong, China), the Arctic University of Norway University of Tromso (Norway), the White Sea Marine Biological Station of St. Petersburg State University (Russia), the Scotland Association of Marine Science (Oban, Scotland, UK) and Interuniversity Institute for Marine Sciences (Eilat, Israel) were informed about the snails sampling. Collection and animal numbers were approved by those authorities.

No special approval could be obtained for sample collection at the site near Cancale (France) because it is not an area of any national park or a private territory, so there was no appropriate authority to apply for any permission.

## Data Availability

A table containing protein spots presence/absence in samples of 10 Littorinid species is available in the Supplemental Files.

## Supplemental Information

Supplemental information for this article can be found online at http://dx.doi.org/10.7717/peerj.8546#supplemental-information.

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
