# Peer review of "Proteomic similarity of the Littorinid snails in the evolutionary context"

_PeerJ, doi:10.7717/peerj.8546_

## Round 0.1 · original submission · Major Revisions

You have presented an interesting study, that is, unfortunately, not clearly written, and therefore, may not reach non-specialist audience. If you expand description of the methodology, the paper will be of interest to a much wider circle of scientists. The reviewers provided useful lists of suggested improvements. I encourage you to take all of them into account, revise and resubmit your manuscript.

Reviewer 1 ·

Basic reporting

The authors present an interdisciplinary study, bridging the fields of proteomics and evolutionary genetics, to study the evolution of several species from the gastropod family Littorinidae. The manuscript is clearly structured according to PeerJ standards, with a well-written introduction that provides the necessary background. While I cannot comment on the literature on proteomics and the study taxa, the remaining literature is well referenced and relevant.

I have two comments to improve the manuscript with respect to basic reporting:
1) While the figures are relevant and of high quality, Fig. 1A and B should be improved with respect to the placement of statistical support values on the trees. Specifically, support values are not clearly placed at their respective branch or seem to be missing (e.g. the posterior probability for the branch leading to L. arcana / L. compressa in Fig. 1B). Moreover, the manuscript would be easier to follow for readers who are not familiar with the study taxa if you would add labels for genera and subgenera to the trees (Fig. 1A and B, Fig. 2).
2) The English language is overall clear and unambiguous, but the manuscript would benefit from improving the grammar, for example in lines 35, 73, 98-101.

Further, I have a few minor suggestions to improve the text:
- Lines 382-388: Moving this section to line 376 would make the whole paragraph easier to follow for the readers.
- Line 309: L. marisrubri should be L. melanostoma.
- Line 491: You refer to Fig. 1, but this should be Fig. 2.

Experimental design

The authors present original primary research, combining the analysis of newly generated proteomics data with already published DNA sequencing data. The research question is well defined and relevant to the field of evolutionary biology and the authors make clear that their research fills a knowledge gap regarding the evolutionary history of littorinid snails.

I commend the authors on the overall study design of comparing proteomics with DNA sequencing data analyses. It is great to see the use of statistical similarity measures to compare trees estimated from proteome and DNA sequencing data. However, the chosen methods are solely based on similar clustering or pairwise distances between tips of the trees, and none of the methods take tree topologies or branch lengths into account. I would like to refer you therefore to the following paper that evaluates the performances of different tree distance measures to choose an additional statistical measure: https://academic.oup.com/sysbio/article/64/2/205/1630737


Further, I have the following minor suggestions how to improve the methods descriptions:
1) Line 238-241 (and throughout the manuscript and figures): For readers who are unfamiliar with littorinid snails it is sometimes difficult to follow which species belong to which genus or subgenus, as not all species names used in the figures are used consistently throughout the manuscript. In this particular sentence, it is not clear which species belong to Littorina (Neritrema) and Littorina (Neritrema / Littorina).
2) Line 253: Please refer to the supplement 1 where the parameters are described more in detail.

Validity of the findings

The authors have provided all underlying data. Newly generated presence/absence proteomics data has been provided as supplemental material and the sources of the analysed published data are available. While I cannot review the quality of the proteomics data, I have a few comments on the interpretation and discussion of results obtained from DNA sequencing data.

The DNA sequencing data analysed in this manuscript consists of two mitochondrial and one nuclear locus. I am missing a discussion on the possibility that the phylogenetic tree based on these loci does not correspond to the species tree (or genome-wide tree), which is the basis to the discussion of the findings from proteomics data. For example, in lines 402-404 you mention that L. obtusata and L. fabilis were shown to share mitochondrial haplotypes. Such a pattern might have been caused by hybridization which could have affected the topology of the tree based on DNA sequencing data.

Also, you mention in line 287 that low statistical support comes from unstable clustering pattern. I am missing some further discussion of this finding, especially keeping in mind that the phylogenetic analysis of DNA sequencing data was only based on three markers which might not contain enough substitutions to resolve this part of the tree. Alternatively, gene flow might explain the low posterior probabilities at these branches (see above).

Further, I noticed that the two Littoraria species are clustering as sister group to the two Echinolittorina species in the DNA sequencing data, but as sister group to all Littorina species in the proteomics tree. This is only mentioned in lines 303-304, but missing from lines 276-277 and other sections. I would suggest to discuss this difference between the two topologies.

Reviewer 2 ·

Basic reporting

I read this manuscript with great interest. I am a molecular systematist, specialised on working with DNA sequences, but I am always open to other forms data that can inform us about relationships of particularly closely related taxa. As such, I wanted to be convinced that proteomics could be an interesting source of such information. As DNA sequence data is what the majority of researchers are using, I am sure I am not alone in this.

Unfortunately I was not convinced, rather I was left somewhat confused by what the data actually is and how it was used. Keep in mind that I (along with the majority of the readers of an article such as this) have little experience with proteomics, how the data is collected, and what it actually is. I elaborate on this below.

Experimental design

The research question is clear, the potential results interesting, but the reader needs to be convinced, is this an approach that I would like to use?

The authors state that their approach is comparable to AFLP, RFLP, RAPD, etc. Fine, but what is the data actually? Table S0 suggests that there are almost 800 characters for which there is presence/absence data. But looking at the names of the characters, it would seem that for many characters we are actually talking about different forms of the same protein (Ah1, Ah2, Ah3 etc). Is that right? If so, I cannot find anywhere how many individual proteins were used to collect data. Surely much less than 796. Does coding each form of a protein as an independent character have an effect on results? What if each protein was coded as a multistate character? Maybe such things have been discussed in the literature earlier, but as I said, I am unfamiliar with that.

A couple of hundred proteins must be a small fraction of all the proteins found in the specimens at the time of sampling, how have exactly these proteins been chosen? Again, you need to convince somebody who is frustrated with DNA data and is looking for another source of phylogenetic information for their problem, but is not an expert in the field of proteomics.

For phylogenetic analysis, the dataset is reduced to a distance matrix based on the Jaccard index. This is then made into a dendrogram using UPGMA. Any particular reason why UPGMA is used? Why not a neighbour joining algorithm? Could UPGMA be the reason why E.mil+E.mar do not come out as sister to L.mel+L.ard?

For the molecular analysis, the authors state that MrBayes is used in a standard fashion, yet they show an ultrametric tree. MrBayes does not give an ultrametric tree under standard analyses. How have the authors made the tree ultrametric? This is not clear at all from the text.

Validity of the findings

The authors compare their proteomic data to DNA sequence data from three loci, totally unrelated to the proteins. My initial thought was, what if we had the DNA sequences for each of the proteins used in the study, would they be as informative as the proteomic data? Now it is not really a fair comparison, unless the point is to say that one can relatively cheaply do the proteomic analysis as suggested by the authors, compared to doing a whole genome sequence to find the DNA sequence of the same proteins. Can the authors comment on this in the manuscript?

Additional comments

In general, I found the study to be interesting, but the manuscript should be revised to make it clearer why such data could be useful for others. Write it for the ignorant (like me), this study is not aimed at specialists in proteomics, it is aimed at the systematist interested in their organisms and their relationships.

---

## Round 0.2 · Minor Revisions

Thank you very much for making significant improvements to the manuscript. At this stage, I request that your work on improving the quality of written English and making sure that the list of references in comprehensive. One of the reviewers has provided an annotated manuscript that may help you with this task. Happy holidays!

Reviewer 1 ·

Basic reporting

The language in the article has improved very much compared to the previous version. The authors will find minor language corrections in the attached PDF to further improve the language.

Also, I have made suggestions for population genomics, phylogenetics and phylogenomics literature to be cited in the attached PDF.

The figures have been improved with respect to the placement of statistical support values and taxon labeling. However, the resolution is not ideal and should be increased.

Experimental design

I only have one minor comment on the methods section, where the parameters used in the phylogenetic analysis using MrBayes should be included (see PDF).

Validity of the findings

No comment

Additional comments

I commend the authors for the improvements to the article compared to the previous version. I attached a combined PDF of the main document and the supplementary material with minor comments to the article.

Annotated reviews are not available for download in order to protect the identity of reviewers who chose to remain anonymous.

Reviewer 2 ·

Basic reporting

I thank the authors for their revision, which makes the manuscript now much clearer, I enjoyed reading it! And it has intrigued me about using proteomics to look at closely related species and species boundaries.

Experimental design

All looks good now.

Validity of the findings

The revised manuscript is now much clearer on the use of proteomics for looking at species level questions.

---

## Round 0.3 · accepted · Accept

Thank you very much for making all the requested changes. You have really improved the manuscript. I recommend it for acceptance.